# Decision makers consider all options in choice triplets

**Douglas G. Lee** [1,2]*

**1** School of Electrical and Electronic Engineering, University College Dublin, Dublin, Ireland, **2** Paris Brain Institute, Paris, France

* DouglasGLee@gmail.com

## Abstract

Most contemporary decision-making research focuses on choices between only two alternative options, in spite of the fact that most real-world decisions involve more than two options. Beyond this practical point, multi-option decisions are also important from a theoretical perspective. Experimental and computational studies have demonstrated that the composition of a set of choice options has predictable effects on choice outcomes. Specifically, with more options available to choose from, responses are slower and more stochastic. This effect is amplified when the values of the options (including the worst option in the set) are more similar to each other. In this study, we provide further evidence of these known effects. We also provide evidence that metacognitive factors such as feelings of confidence in the response or mental effort exertion during deliberation show similar effects as the cognitive factors (consistency between choices and value estimates, response speed). Finally, we provide novel evidence that value estimates are refined during deliberation for all options in choice triplets, similar to what has previously been show for choice pairs.

**Data availability statement:** The data and analysis script for this study are publicly available at https://osf.io/d4urc/.

## Introduction

Most decisions that we face in our lives include several different options from which we are free to choose our preferred. This is true for a variety of domains, including consumer decisions (e.g., which smartphone to buy, which apartment to rent), financial decisions (e.g., which stock portfolio to invest in, which insurance policy to enroll in), and even social decisions (e.g., which mate to pursue, which colleague to mingle with). In the modern age, for most decisions, there are an increasing number and variety of options, as well as an increasing technical feasibility of realistically considering the multitude of potential options. While this obviously offers great advantages to decision makers (DMs), it simultaneously offers some important disadvantages.

For one, the size of the set of options can overwhelm DMs with more information than they can efficiently process, potentially leading to suboptimal choice behavior. Published evidence of this sort of effect dates back to Hick [1], who noted that people take longer to respond when they are presented with more options to choose from. Known as Hick's law, this phenomenon suggests that the presence of more options causes greater choice difficulty or uncertainty, which can lead to lower accuracy. Indeed, it is known from empirical evidence

**Funding:** The author(s) received no specific funding for this work.

**Competing interests:** The authors have declared that no competing interests exist.

that choices are more stochastic when the choice set contains more elements [2–5]. This could be interpreted under the theory of divisive normalization, which holds that neural activity representing the value of each option is normalized by the sum of the values of all options in a choice set [6]. In this way, if DMs were presented with many options to consider, their neural value representations would be sparse and, therefore, the precision of such representations would be low. It would then directly follow that choices between options represented in such a way would be more stochastic.

The literature on so-called "choice overload" shows that when the number of choice options increases, the level of choice confidence or satisfaction that DMs have decreases and the level of regret increases (see [7] for a review). Evidence of choice overload has been supported by neuroimaging showing that neural activity in regions of the brain known to encode value (striatum and anterior cingulate cortex) initially increased with choice set size, but then decreased as choice set size increased further [8]. However, behavioral findings related to choice overload have been mixed [9,10].

The specific composition of a set of options can also have unexpected effects on choice behavior, with inconsistencies arising according to decision context. One frequent observation is that when a third option is added to a choice set containing a pair of other options, choice stochasticity increases. Moreover, choice stochasticity increases as a function of the value of the third option (even when it is always strictly lower in value than the other two options). This is explainable by divisive normalization [6], but also by theories of attention. For example, as the value of the third option (often referred to as a "distractor") increases, it might capture more attention from the other options, thus causing interference in the comparison between the first and second options [11].

Beyond choice stochasticity, the availability of more alternative options could delay responses and decrease choice confidence. Metacognitive variables such as confidence are thought to be important to guide decisions, including both the current decision [12] and future decisions [13,14]. According to the metacognitive control of decisions [15] theory, confidence is actually what determines when deliberation will end and a choice will be made. In brief, under this theory, the amount of cognitive resources deployed during deliberation is controlled by an effort-confidence tradeoff, which relies on a proactive anticipation of how deliberation might perturb the internal representations of subjective value in a way that generally increases confidence in the choice. This approach predicts that choice stochasticity, confidence, response time (RT), and subjective feelings of mental effort exertion will all be impacted by the values (in particular, the differences in the values) of all available options in choice sets of any size.

Most contemporary research on simple value-based decision-making involves choices between *pairs* of options (but see the specific literature on *context effects* in decision-making; [16–18]). Clearly, understanding situations where DMs must choose from among more than two options is also important. In this current work, we present evidence related to choice confidence and mental effort – in addition to the classical choice variables of accuracy/consistency and RT – in a task where each choice set contains three options. We test for relationships between these behavioral variables and the value estimates of the individual options to determine whether the relationships in trinary choice data are similar to those in binary choice data. We also specifically test for the impact of the worst option within a triplet, to determine whether all options are considered during choice deliberation or only the best two alternatives for each decision. Our results reaffirm previous empirical findings (i.e., choices are more stochastic and slower when the value of the worst option is greater) and provide additional evidence that metacognitive factors mirror cognitive factors (i.e., feelings of confidence are lower and feelings of effort are higher when the value of the worst option is greater).

Additionally, we investigate a separate phenomenon related to choice that has been well-documented over the decades: choice-induced preference change or *value refinement*. This phenomenon demonstrates that DMs tend to change their value estimates after they are asked to choose between options that they initially valued similarly ([15,19–26]; see [27] for a meta-analysis). On average, DMs increase their valuations of options that they choose and decrease their valuations of options that they reject. Most decision-making theories do not consider dynamic value refinement during decisions, but there is some recent work in this direction [15,28–30]. Recent work supports value refinement over alternative explanations [31]. In this current study, we test for relationships between the initial value estimates of the individual options and evidence of value refinement. This has already been done repeatedly in binary choice, but never before in trinary choice.

In summary, the primary purpose of this study is to demonstrate that choice behavior is affected by all options within a choice set. Specifically, we seek to demonstrate that choices are facilitated (i.e., they are more consistent with value estimates, faster, and associated with feelings of greater confidence and lesser mental effort) when the differences in the value estimates of all available options are greater and when the average value across available options is greater. A secondary purpose of this study is to demonstrate that people revise their value estimates for all options during choice deliberation. The point is to show that all options are taken into consideration, not simply the best or second-best options.

## Methods

We conducted a behavioral experiment where participants provided initial value estimates and certainty ratings (about the value estimates) for a variety of snack food options, then chose among triplets of options (with each option appearing in only one triplet), then provided final value estimates and certainty ratings for the same options. Prior to these core tasks, participants experienced an exposure phase, during which each option was briefly displayed and participants simply observed the images. The experiment was developed using Matlab and PsychToolbox. Written instructions provided detailed information about the sequence of core tasks within the experiment, the mechanics of how participants would perform the tasks, and images illustrating what a typical screen within each task section would look like. The study was conducted according to the principles expressed in the Declaration of Helsinki. The experiment was approved (opinion number 16–333) by the ethics evaluation committee of lnserm (IORG0003254, FWA00005831), the Institutional Review Board (IRB00003888) of the French Institute of Medical Research and Health.

## Participants

A total of 50 people (39 females, 11 males; age: mean=27, stdev=5, min=19, max=40) participated in this study. These participants were recruited between March 15, 2019, and April 30, 2019, via the internal recruiting system of the Paris Brain Institute, without filtering for gender balance. The experiment lasted approximately 2 hours, and each participant was paid a flat rate of 24€ as compensation for participating. One participant was excluded from our analysis because she was unable to complete the experiment. Three other participants were excluded from our analysis for not performing the tasks properly. All participants were native French speakers, and the experiment was conducted entirely in French. All participants were from a non-patient population with no reported history of psychiatric or neurological illness. All participants provided written informed consent prior to commencing the experiment.

## Materials

The stimuli for this experiment were 300 digital images, each representing a distinct snack food item (fruits, vegetables, nuts, meats, cheeses, chips, crackers, cookies, cakes, pies, and a variety of other suitable snacks).

## Experimental procedure

Prior to commencing the testing session of the experiment, participants underwent a brief training session. The training tasks were identical to the experimental tasks, except that different stimuli (images of beverages) were used.

The experiment itself began with an initial exposure section where all individual items were displayed in a random sequence for 700ms each, to familiarize the participants with the set of options they would later need to consider and to allow them to form an impression of the range of subjective value across all possible options. The main experiment was divided into three sections, following the classic *free-choice paradigm* protocol [32]: Rating, Choice, Rating. There was no time limit for the overall experiment, nor for the different sections, nor for the individual trials. Within-trial event sequences are described below (see Fig 1 for an illustrative example).

**Rating**: Participants were asked to evaluate each item in terms of its subjective desirability. The items were presented one at a time in a random sequence (pseudo-randomized across participants). At the onset of each trial, a fixation cross appeared at the center of the screen for 750ms. Next, a solitary image of a food item appeared at the center of the screen. Participants responded to the question, "How much do you like this item?" using a horizontal slider scale to indicate their value rating for the item (from "I hate it!" to "I love it!", with "I don't care." at the center of the scale). The granularity of the scale was such that 100 different rating

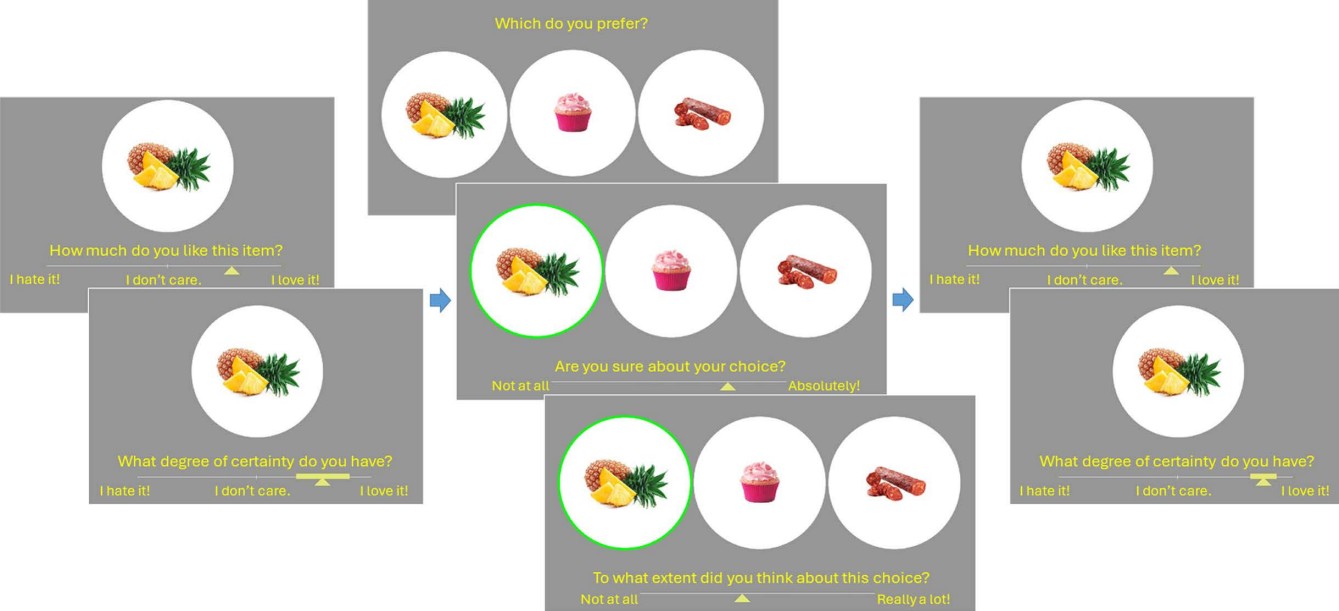

**Fig 1. Core Experimental Tasks.** Participants individually estimated their subjective value of each option, along with their certainty about each estimation; next, they chose their preferred option from triplets of options and provided their subjective confidence and effort estimates for each choice; finally, participants again estimated their subjective value and certainty for each option.

levels were available, although participants were not aware of that. The initial position of the response cursor on each trial was at the neutral center point. Participants then responded to the question, "What degree of certainty do you have?" by expanding a solid bar symmetrically around the cursor of the slider scale to indicate their uncertainty about the value rating they had just provided. They had been instructed and trained to understand that the width of the bar represented their degree of *uncertainty*, as if their true rating might be found anywhere within the range of the solid bar.

**Choice**: Participants were asked to choose among triplets of items in terms of which item they preferred. The entire set of items was presented one triplet at a time in a random sequence. Each item appeared in only one triplet, yielding a total of 100 choices. The triplets were created for each participant such that the subjective rating difference between the best and second-best options and the subjective rating difference between the second-best and worst options were low, medium, or high (orthogonalized across trials). At the onset of each trial, a fixation cross appeared at the center of the screen for 750ms. Next, three images of snack items appeared on the screen: one towards the left, one in the center, and one towards the right. Participants responded to the question, "Which do you prefer?" using the left, down, or right arrow key. Participants then responded to the question, "Are you sure about your choice?" using a horizontal slider scale (from "Not at all" to "Absolutely"). This provided a subjective measure of choice confidence. Finally, participants responded to the question, "To what extent did you think about this choice?" using a horizontal slider scale (from "Not at all" to "Really a lot"). This provided a subjective measure of mental effort expenditure to supplement the objective effort proxy of response time. The granularity of the scales was such that 100 different confidence and effort levels were available, although participants were not aware of that. The initial position of the response cursors on each trial was at the minimum point.

## Data analysis

For all analyses, we removed trials with outlier RT. We defined an outlier as any log-transformed RT more than three median average deviations (MAD) above or below the median log-transformed RT (determined separately for each participant). This resulted in the exclusion of 81 trials (74 too slow, 7 too fast; less than 2% of the 4,600 trials pooled across participants).

For our model-based analyses, we utilized mixed-effects regression models to test for fixed effects of the relevant decision variables. We used the *fitlme* function in Matlab. In all regression models, we included participants as random effects (both intercepts and slopes).

## Results

We report here the relationships of pre-choice value estimates with consistency, confidence, RT, effort, and value refinement. We start with a simple model-free analysis and then continue with regression analysis in the next subsection.

First, we tested how choice consistency related to the difference between the value estimates of the best and 2nd best options (hereafter dVa). Consistency refers to choices that are congruent with preferences inferred from pre-choice value ratings. The dVa variable corresponds to the classical "ease" or inverse difficulty measure in two-alternative forced-choice paradigms. We pooled the data across all participants, then separated it into nine bins of equal size based on dVa (which had a potential range of 0 to 0.99). The experimental design was such that the trials would most likely contain measures of dVa between 0 and 0.11 (52% of trials), between 0.11 and 0.22 (27% of trials), or between 0.22 and 0.33 (17% of trials). We neglect the 4% of trials for which dVa was greater than 0.33 for this preliminary set of analyses.

As expected, consistency was greater for bins of greater dVa (bin1 = 54%, bin2 = 74%, bin3 = 84%).

We then tested how consistency related to the difference between the value estimates of the 2nd best and worst options (hereafter dVb). We performed the same analysis as above, only this time splitting the data into nine bins of equal size based on dVb (which had a potential range of 0 to 0.99). As with dVa, the experimental design was such that the trials would most likely contain measures of dVb between 0 and 0.11 (51% of trials), between 0.11 and 0.22 (28% of trials), or between 0.22 and 0.33 (18% of trials). Importantly, dVa and dVb were orthogonally manipulated. We neglect the 3% of trials for which dVb was greater than 0.33 for these preliminary analyses. We found that dVb had a clear impact on consistency, qualitatively similar to (but quantitatively lesser than) the impact of dVa (bin1 = 61%, bin2 = 70%, bin3 = 74%). Fig 2 summarizes these effects by showing results across nine distinct bins formed by crossing three levels of dVa (0–0.11 or low, 0.11–0.22 or mid, and 0.22–0.33 high) with three levels of dVb (0–0.11 or low, 0.11–0.22 or mid, and 0.22–0.33 high).

Second, we tested how choice confidence related to dVa. We pooled the data across all participants, then separated it into bins of equal size based on dVa. As expected, confidence was greater for bins of greater dVa (bin1 = 62%, bin2 = 70%, bin3 = 74%). We then tested how confidence related to dVb. We repeated the same analysis, only this time splitting the data into bins of equal size based on dVb. We found that dVb had a clear impact on confidence, qualitatively similar to (but quantitatively lesser than) the impact of dVa (bin1 = 63%, bin2 = 70%, bin3 = 74%). Fig 3 summarizes these effects by showing results across nine distinct bins formed by crossing three levels of dVa with three levels of dVb.

Third, we tested how response time (RT) related to dVa. We pooled the data across all participants, then separated it into bins of equal size based on dVa. As expected, RT was lesser for bins of greater dVa (bin1 = 3.1s, bin2 = 2.6s, bin3 = 2.2s). We then tested how RT related to dVb. We repeated the same analysis, only this time splitting the data into bins of equal size based on dVb. We found that dVb had a clear impact on RT, qualitatively similar to (but quantitatively lesser than) the impact of dVa (bin1 = 3.0s, bin2 = 2.6s, bin3 = 2.3s). Fig 4

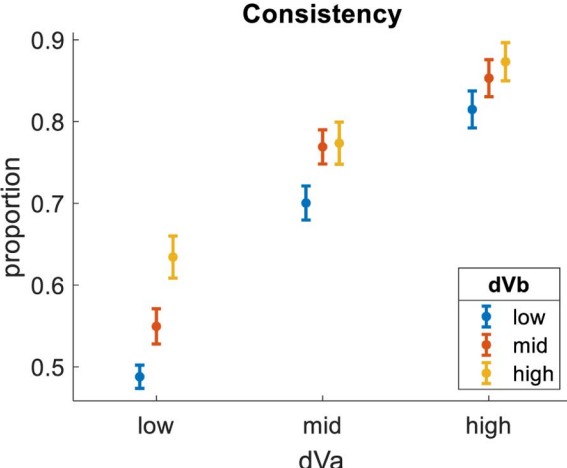

**Fig 2. Impact of dVa and dVb on Consistency.** dVa is the value difference between the highest-rated and 2nd-highest-rated options in each choice triplet. dVb is the value difference between the 2nd-highest-rated and lowest-rated options in each choice triplet. Consistency is defined as the highest-rated option in a triplet being chosen. Consistency is greater for trials with greater dVa and trials with greater dVb. Dots represent means (trials pooled across participants). Error bars represent S.E.M.

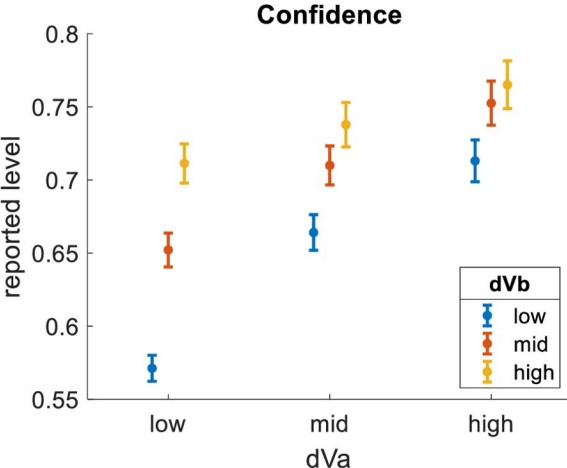

**Fig 3. Impact of dVa and dVb on Confidence.** dVa is the value difference between the highest-rated and 2nd-highest-rated options in each choice triplet. dVb is the value difference between the 2nd-highest-rated and lowest-rated options in each choice triplet. Confidence is greater for trials with greater dVa and trials with greater dVb. Dots represent means (trials pooled across participants). Error bars represent S.E.M.

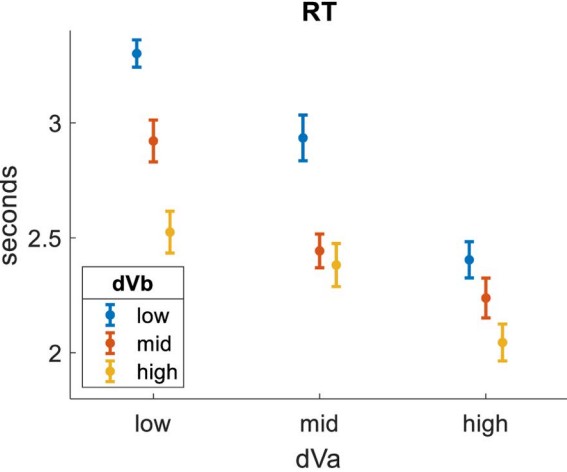

**Fig 4. Impact of dVa and dVb on RT.** dVa is the value difference between the highest-rated and 2nd-highest-rated options in each choice triplet. dVb is the value difference between the 2nd-highest-rated and lowest-rated options in each choice triplet. RT is lesser for trials with greater dVa and trials with greater dVb. Dots represent means (trials pooled across participants). Error bars represent S.E.M.

summarizes these effects by showing results across nine distinct bins formed by crossing three levels of dVa with three levels of dVb.

Fourth, we tested how subjective metal effort related to dVa. We pooled the data across all participants, then separated it into bins of equal size based on dVa. As expected, effort was lesser for bins of greater dVa (bin1 = 39%, bin2 = 33%, bin3 = 29%). We then tested how effort related to dVb. We repeated the same analysis, only this time splitting the data into bins of equal size based on dVb. We found that dVb had a clear impact on effort, qualitatively similar to (but quantitatively lesser than) the impact of dVa (bin1 = 37%, bin2 = 34%, bin3 = 31%).

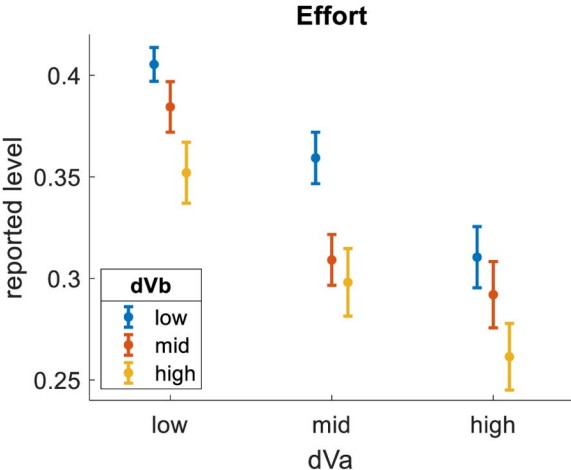

**Fig 5. Impact of dVa and dVb on Effort.** dVa is the value difference between the highest-rated and 2nd-highest-rated options in each choice triplet. dVb is the value difference between the 2nd-highest-rated and lowest-rated options in each choice triplet. Effort is lesser for trials with greater dVa and trials with greater dVb. Dots represent means (trials pooled across participants). Error bars represent S.E.M.

Fig 5 summarizes these effects by showing results across nine distinct bins formed by crossing three levels of dVa with three levels of dVb.

Finally, we tested how value refinement related to dVa. Value refinement refers to how value estimates change from pre- to post-choice ratings and is thought to relate to mental effort and information processing during choice deliberation [15,21]. Here, we examined the spreading of alternatives (SoA; [32]), which is the post- minus pre-choice value difference between the chosen and unchosen options (here we averaged the values of the two unchosen options in each triplet). This measure supports the idea that values are refined during deliberation in order to facilitate the choice, typically resulting in increased ratings for chosen items and decreased ratings for unchosen items. As expected, SoA was lesser for bins of greater dVa (bin1 = 8%, bin2 = 5%, bin3 = 3%). We then tested how SoA related to dVb. We repeated the same analysis, only this time splitting the data into bins of equal size based on dVb. We found that dVb had a clear impact on SoA, qualitatively similar to (but quantitatively lesser than) the impact of dVa (bin1 = 7%, bin2 = 6%, bin3 = 4%). Fig 6 summarizes these effects by showing results across nine distinct bins formed by crossing three levels of dVa with three levels of dVb.

### Regression analysis

To better quantify the effects reported above, we conducted a series of mixed effects regression analyses. We separately regressed consistency, confidence, log(RT), effort, and SoA (logistic regression for consistency, linear regression for all other dependent variables) on dVa and dVb. In line with the model-free results reported above, all fixed effects beta coefficients were statistically significant (see Table 1).

To more closely examine the impact of the values of each option within a choice triplet (best, middle, and worst), we repeated the above regression analyses this time replacing the independent variables with the value of the best option ($V_{best}$), the value of the middle option ($V_{mid}$), and the value of the worst option ($V_{worst}$) on each choice trial. The results show that all options impact each of the choice variables (see Fig 7). In particular, higher values for the best

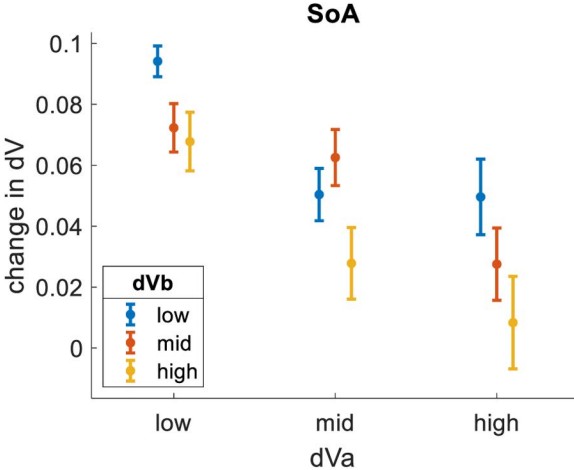

**Fig 6. Impact of dVa and dVb on SoA.** dVa is the value difference between the highest-rated and 2nd-highest-rated options in each choice triplet. dVb is the value difference between the 2nd-highest-rated and lowest-rated options in each choice triplet. SoA is the post- minus pre-choice value difference between the chosen and unchosen options (averaged). SoA is lesser for trials with greater dVa and trials with greater dVb. Dots represent means (trials pooled across participants). Error bars represent S.E.M.

**Table 1. Relationship between value differences and choice behavior.**

| Fixed effects coefficients | | | | | |
|---|---|---|---|---|---|
| | **Consistency** | **Confidence** | **log(RT)** | **Effort** | **SoA** |
| *difference in value: best - mid (dVa)* | 0.92 ($p < .001$) | 0.49 ($p < .001$) | -0.89 ($p < .001$) | -0.39 ($p < .001$) | -0.21 ($p < .001$) |
| *difference in value: mid - worst (dVb)* | 0.45 ($p < .001$) | 0.37 ($p < .001$) | -0.75 ($p < .001$) | -0.23 ($p < .001$) | -0.16 ($p < .001$) |
| $r^2$ | 0.10 | 0.24 | 0.34 | 0.31 | 0.06 |

option seem to facilitate the choice (greater consistency and confidence; lesser RT, effort, and SoA), whereas higher values for either of the other options seem to impede the choice.

The inclusion of all individual value ratings (best, mid, worst) generally resulted in more explained variance (i.e., greater $r^2$) relative to the models based on dVa and dVb. One difference between the two sets of regressors is that information about the sum of the values is also available in the models with all individual ratings, in addition to the information about the differences between the values (which is available in both sets of models). With that in mind, we repeated the first set of regression analyses, this time including the average of the values in a choice set (V; best + mid + worst, divided by three for normalization) as an additional explanatory variable. Interestingly, V was statistically significant in all models and seems to facilitate the choice ([Fig 8]).

## Value refinement occurs for all options

The results that we reported above demonstrate the relationship between the values of the individual options within a choice triplet and value refinement in the form of the spreading of alternatives (SoA; see [Fig 7E]). Note that we defined SoA as the rating change for the value of the chosen option minus the *average* rating change for the values of both unchosen options. We did this because SoA in the contexts of choice triplets has never been examined before, so we had to determine a reasonable adjustment to the classic definition. However, our definition does not allow us to observe whether the value of the

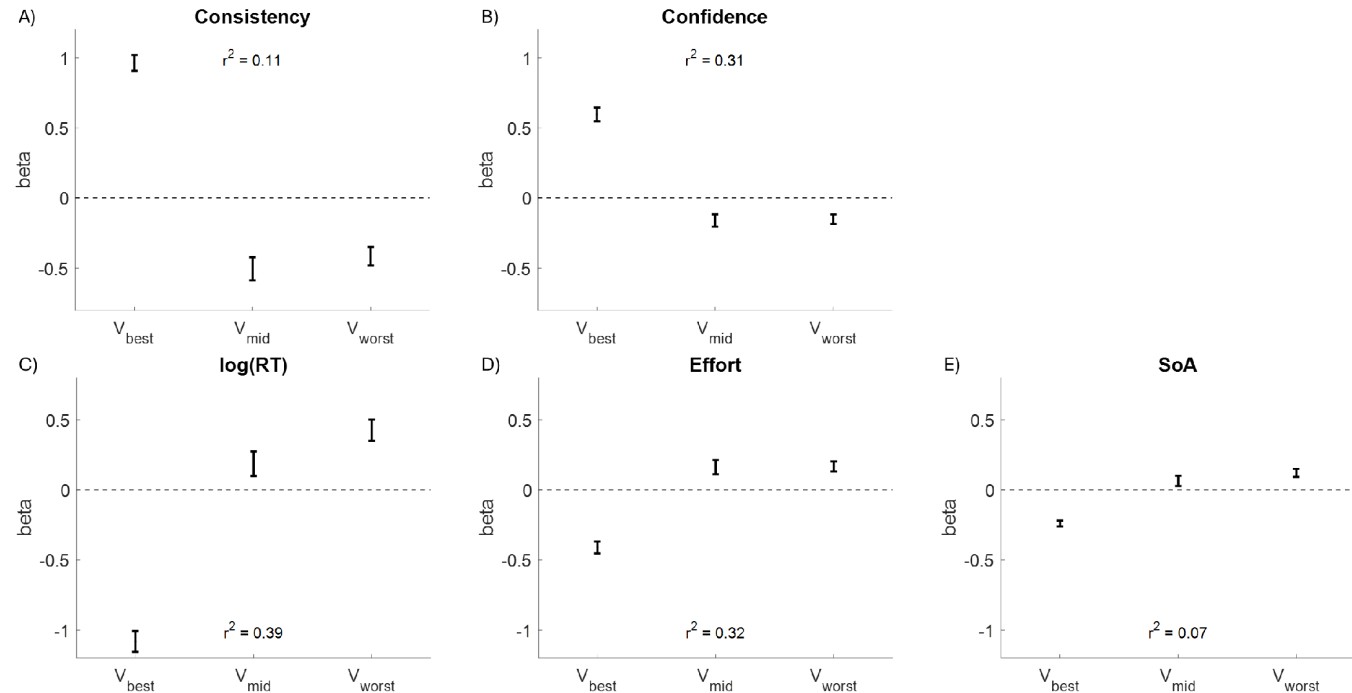

**Fig 7. Fixed Effects Coefficient Estimates.** Regressing consistency (plot A), confidence (plot B), log(RT) (plot C), effort (plot D), and the spreading of alternatives (SoA; plot E) on the value of the best option ($V_{best}$), the value of the middle option ($V_{mid}$), and the value of the worst option ($V_{worst}$), coefficients for all dependent variables indicate that $V_{best}$ facilitates the choice (positive beta for consistency and confidence, negative beta for log(RT), effort, and SoA) and that $V_{mid}$ and $V_{worst}$ impede the choice (negative beta for consistency and confidence, positive beta for log(RT), effort, and SoA). Error bars represent estimates +/- standard errors.

worst option changes in a systematic way, or if the entire SoA effect is the same as what has been reported many times in the past (i.e., the rating change of the chosen option minus the rating change of the next-best option). To better understand our data, we first calculated the average rating change across all trials for the individual options (chosen: *best*; better unchosen: *mid*; worse unchosen: *worst*). Chosen options increased by an average of 4% ($p$ <.001), better unchosen options decreased by an average of 4% ($p$ <.001), and worse unchosen options did not change ($p$ =.477; see Fig 9A). This confirms previous results with respect to the chosen and next-best options and suggests that perhaps value estimates for the worst options are not refined during deliberation. We then regressed the rating change for the individual options ($dR_{best}$, $dR_{mid}$, $dR_{worst}$) separately on the values of each option ($V_{best}$, $V_{mid}$, $V_{worst}$). The beta coefficients for $dR_{best}$ and $dR_{mid}$ align with previously reported results (Fig 9B, C): $dR_{best}$ had a negative relationship with $V_{best}$ (beta = -0.31, $p$ <.001), a positive relationship with $V_{mid}$ (beta = 0.11, $p$ <.001), and no relationship with $V_{worst}$ ($p$ =.590); $dR_{mid}$ had a positive relationship with $V_{best}$ (0.09, $p$ <.001), a negative relationship with $V_{mid}$ (-0.25, $p$ <.001), and no relationship with $V_{worst}$ ($p$ =.696). With respect to the value of the worst option (Fig 9D), $dR_{worst}$ had a positive relationship with $V_{best}$ (beta = 0.05, $p$ <.001), no relationship with $V_{mid}$ ($p$ =.658), and a negative relationship with $V_{worst}$ (beta = -0.19, $p$ <.001). Interestingly, the pattern for the worst option is similar to the pattern for the middle option, in the sense that its value rating decreases more when it is initially greater, but also that its value increases more (or decreases less) when the best option has a greater initial value. So, whereas the value of the worst option does

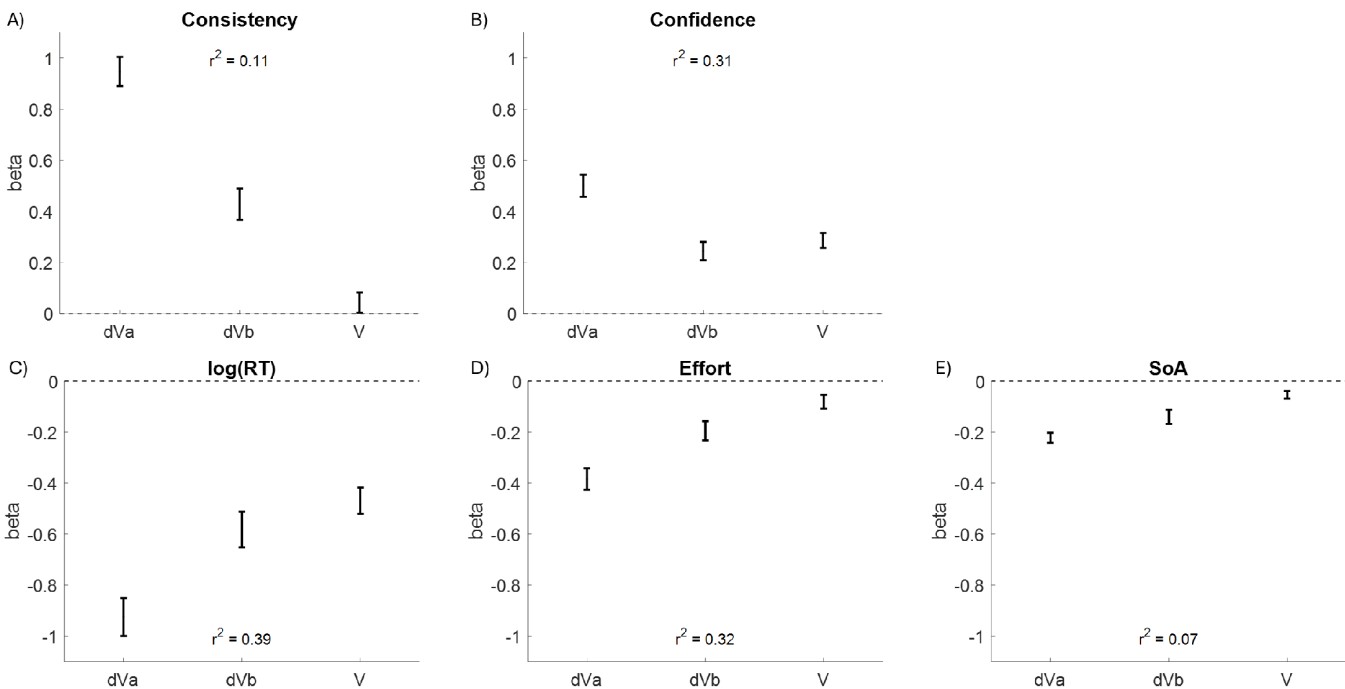

**Fig 8. Fixed Effects Coefficient Estimates.** Regressing consistency (plot A), confidence (plot B), log(RT) (plot C), effort (plot D), and the spreading of alternatives (SoA; plot E) on the difference in value between the best and middle options (dVa), the difference in value between the middle and worst options (dVb), and the value average across options (V): of note, all coefficients for V indicate that it facilitates the choice (positive beta for consistency and confidence, negative beta for log(RT), effort, and SoA). Error bars represent estimates +/- standard errors.

not change on average across all trials, it does change in a context-dependent manner on a trial-by-trial basis.

### Feelings of certainty influence choice

Previous studies have shown that choices can be facilitated by the subjective feelings of certainty that participants have about their value estimates for the options [15,20–22,33–35]. We thus tested for similar findings in our data. We separately regressed consistency, confidence, log(RT), effort, and SoA on the difference in value between the best and middle options (dVa), the difference in value between the middle and worst options (dVb), the average value of the choice set (V), and the average value certainty of the choice set (C). The results only partially replicate previous findings. In particular: certainty had a positive relationship with confidence, a negative relationship with SoA, and a negative trend with log(RT), but no significant relationship with consistency or effort (see Table 2).

We also tested how changes in individual option evaluations (ignoring the choice data) related to value certainty. We first regressed the change in certainty (from pre- to post-choice ratings) on initial certainty. The resulting beta coefficient showed a strong negative relationship (beta = -0.82, $p$ <.001). More importantly, there was an increase in certainty on average across all options (intercept = 0.73, $p$ <.001). The theory of value refinement [15] holds that increased deliberation should take place when value certainty is lower, and that this should lead to both greater increases in certainty *and* greater average perturbations in value estimates. We thus regressed the magnitude of the change in value ratings on initial certainty. Similar to the results related to certainty gain, the resulting beta coefficient showed a strong negative relationship (beta = -0.13, $p$ <.001; intercept = 0.23, $p$ <.001).

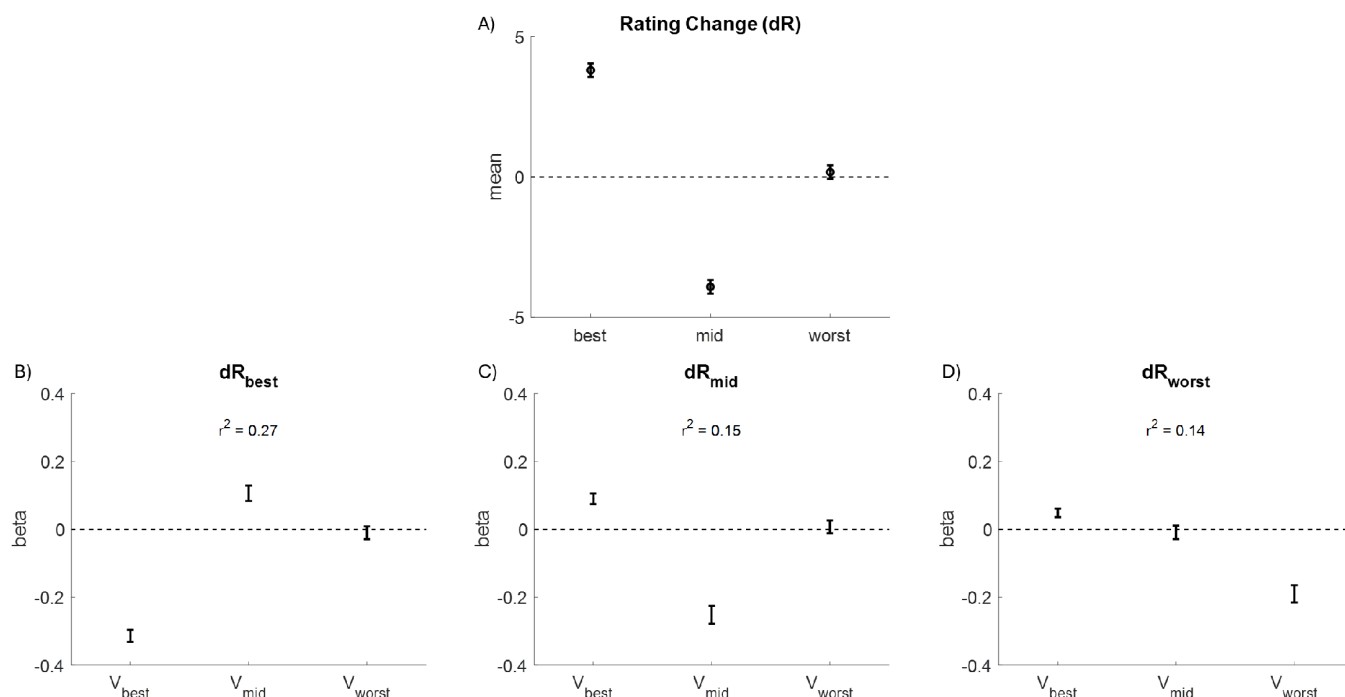

**Fig 9. Value Change as a Function of Initial Values.** A) Value rating change for the chosen (best), better unchosen (mid), and worse chosen (worst) option (averaged across all trials and participants; error bars represent S.E.M.). B-D) Fixed effect beta weights when regressing the value change of the chosen option ($dR_{best}$; plot B), the value change of the better unchosen option ($dR_{mid}$; plot C), and the value change of the worse unchosen option ($dR_{worst}$; plot D) on the initial value of each option ($V_{best}$, $V_{mid}$, $V_{worst}$). Error bars represent estimates +/- standard errors.

**Table 2. Relationships between value plus certainty and choice behavior.**

| Fixed effects coefficients | | | | | |
|---|---|---|---|---|---|
| | Consistency | Confidence | log(RT) | Effort | SoA |
| *difference in value: best - mid (dVa)* | 0.94 ($p$ <.001) | 0.49 ($p$ <.001) | -0.93 ($p$ <.001) | -0.38 ($p$ <.001) | -0.22 ($p$ <.001) |
| *difference in value: mid - worst (dVb)* | 0.43 ($p$ <.001) | 0.24 ($p$ <.001) | -0.58 ($p$ <.001) | -0.19 ($p$ <.001) | -0.15 ($p$ <.001) |
| *set value (V)* | 0.04 ($p$ =.302) | 0.28 ($p$ <.001) | -0.48 ($p$ <.001) | -0.08 ($p$ =.003) | -0.06 ($p$ <.001) |
| *set certainty (C)* | -0.02 ($p$ =.848) | 0.13 ($p$ =.047) | -0.22 ($p$ =.065) | 0.03 ($p$ =.525) | -0.17 ($p$ <.001) |
| $r^2$ | 0.11 | 0.32 | 0.39 | 0.33 | 0.07 |

## Discussion

In this study, we report experimental results demonstrating relationships between the subjective value estimates of triplets of snack food options and choice consistency, choice confidence, response time (RT), subjective mental effort, and value refinement (measured as the spreading of alternatives or SoA). Specifically, choice behavior was affected by all options within the triplets, not only the better options. Decisions seem to have been facilitated both by a greater difference in the value estimates of the highest- and 2nd-highest-rated options within a triplet as well as by a greater difference in the value estimates of the 2nd-highest- and lowest-rated options within a triplet. We refer to facilitated decisions as those with greater consistency with individual value estimates, greater confidence in having chosen the best option, faster responses, lower feelings of mental effort exertion during deliberation, and lesser refinements of individual value estimates from before to after the decisions. Our findings align

with what has previously been reported in a variety of studies based on theories related to, for example, divisive normalization [6], attention capture [11], or mutual inhibition between response circuits in neural and computational decision networks [36,37]. They also align with the predictions of the metacognitive control of decisions model, which has the added allure of accounting for value refinement [15]. Although we provide evidence that decisions among three options yield similar results to decisions between only two options, future work might seek to determine if there is an option set size at which patterns in the data start to deviate.

Beyond the impact of the differences in value across the options, we also reported a clear and consistent impact of the sum (or rather, the mean) of the option values and choice behavior. It seems that decisions involving options of greater subjective value were facilitated in a manner qualitatively similar to the impact of value difference. Most theories do not predict this effect, although it has been reported before [22,38–41] and does appear as an output of certain computational models [36,42,43]. Effects related to value sum/mean are typically not focused on in the literature, yet they might provide insight into which type of mechanisms are at work during decision-making (c.f., the debate between divisive normalization and attention capture; [44,45]). According to the tenets of divisive normalization, the value representation of each individual option is scaled by the inverse of the sum of all option values in the choice set. This directly implies that the normalized value differences will be smaller when the option set is more valuable, holding constant the actual difference in value estimates. The result would be that decisions would be impeded, not facilitated, when value sum/mean was greater. So, the findings related to value sum/mean go against the theory of divisive normalization in this context. With respect to attentional capture, the prediction would be that more highly-rated options would more strongly attract attention. This would mean that when the option set is more valuable, the attention of a DM would shift more frequently (i.e., fixation durations would be shorter). But it is reasonable to claim that more valuable options would not only attract but also retain attention. This would lead to *longer* fixation durations for more valuable options, which would align with certain theoretical models [46,47]. The tension between maintaining attention on a high-valued option versus shifting attention towards an alternative high-valued option should be explored further in future work.

Finally, we provide additional support for the theory of value refinement [15,21] beyond what has been shown in many previous studies based on two-alternative paradigms. Ours may be the first study to test for value refinement (measured as the spreading of alternatives from pre- to post-choice value ratings or SoA) in choice sets containing three alternatives. We confirmed that SoA occurs in trinary choice similarly to how it occurs in binary choice. More importantly, we showed that the value estimate of the worst option in a triplet also changes as a function of the value of the best option, indicating that information about all options was considered during deliberation. The ratings of the worst options increased more when they were initially rated lower, which could potentially be a regression-to-the-mean effect related to the boundary of the rating scale (i.e., if ratings randomly fluctuate when repeatedly queried, the lowest ratings would be more likely to increase because they could not decrease beyond the limit of the rating scale). Note that the average rating change of the worst option across all trials was not statistically different from zero, which refutes a pure regression-to-the-mean effect. Crucially, the ratings of the worst options increased more when the best options were rated higher. This may indicate that the repulsive effect of choosing between similarly-rated options (i.e., classical SoA) becomes an attractive effect when one option clearly dominates another. Future work might explore the intricacies of value refinement for choices among multiple alternatives.

Certainty about value estimates has previously been suggested to facilitate choices. Specifically, greater value certainty corresponds to reduced deliberation time and lesser value

refinements. Additionally, gains in certainty measured from post- minus pre-choice ratings are greater when pre-choice certainty is lesser. We replicated these previous binary choice findings in our trinary choice study. However, we did not replicate other previous findings that greater value certainty corresponds to greater choice consistency and confidence.

## Conclusion

In summary, our results demonstrate that choice behavior is affected by all options within a choice set. Choices are facilitated by greater differences in the value estimates of all available options. They also seem to be facilitated when the average value across available options is greater. Apart from facilitation, our results support the idea that people revise their value estimates during choice deliberation. This value refinement occurs for all available options.

## Limitations on generalizability

This study examined choices between snack food options. It is possible that the results might have differed if different types of decisions were examined. The group of participants in this study was skewed towards the majority being female. We did not examine gender differences, but it is possible that the results might have differed if the group had gender balance or a male majority. In this study, we only considered a choice set size of three options. There might be a point at which adding more options to a set changes the result that all options are considered during deliberation. Our results did not replicate previous findings that value estimate certainty enhances choice consistency and confidence. It could be that metacognitive variables such as certainty become less impactful when choice set size increases beyond two options.

## Author contributions

**Conceptualization:** Douglas G. Lee.

**Data curation:** Douglas G. Lee.

**Formal analysis:** Douglas G. Lee.

**Investigation:** Douglas G. Lee.

**Methodology:** Douglas G. Lee.

**Project administration:** Douglas G. Lee.

**Supervision:** Douglas G. Lee.

**Validation:** Douglas G. Lee.

**Visualization:** Douglas G. Lee.

**Writing – original draft:** Douglas G. Lee.

**Writing – review & editing:** Douglas G. Lee.

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
