## [Decision Letter · Decision Letter 0]

30 Dec 2024

PONE-D-24-57520Decision Makers Consider All Options in Choice TripletsPLOS ONE

Dear Dr. Lee,

Thank you for submitting your manuscript to PLOS ONE. After careful consideration, we feel that it has merit but does not fully meet PLOS ONE’s publication criteria as it currently stands. Therefore, we invite you to submit a revised version of the manuscript that addresses the points raised during the review process.

We look forward to receiving your revised manuscript.

Kind regards,

Yuanchao Liu

Academic Editor

PLOS ONE

Journal Requirements:

Additional Editor Comments (if provided):

The author should revise this manuscript carefully. 1)The study presents the results of original research in an appropriate fashion. 2)In problem statement. the the justification of conducting the research should be well explained. 3)Methodology and findings are performed to a high technical standard and sufficient detail. but the author should describe more a bout the sampling. 4)The research meets all applicable standards for the ethics of experimentation. 5)Conclusions are presented in a good way but it needs short general conclusion, mentioning some research limitations and implications.

Reviewers' comments:

Reviewer's Responses to Questions

**Comments to the Author**

1. Is the manuscript technically sound, and do the data support the conclusions?

Reviewer #1: No

Reviewer #2: Yes

Reviewer #3: Yes

2. Has the statistical analysis been performed appropriately and rigorously? 

Reviewer #1: No

Reviewer #2: Yes

Reviewer #3: Yes

3. Have the authors made all data underlying the findings in their manuscript fully available?

Reviewer #1: Yes

Reviewer #2: Yes

Reviewer #3: Yes

4. Is the manuscript presented in an intelligible fashion and written in standard English?

Reviewer #1: Yes

Reviewer #2: Yes

Reviewer #3: Yes

5. Review Comments to the Author

Reviewer #1: The author(s) must reconsider to provide a comprehensive conceptual framework in the introduction that can help readers understand this topic better. The conceptual framework can also help the author(s) to refine the theory integration used in the research.

Reviewer #2: In the last paragraph of the problem statement, the justification for conducting the research should be clearly and well stated.

In the methodology, it is stated that the sample size is 50, but 39 women are mentioned, and no explanation is given in this regard.And it is better to explain why women were chosen as the statistical sample.

In the discussion section, it is better to add a brief general conclusion.

The limitations of the research, research suggestions and implications should also be discussed.

Reviewer #3: The study work is meaning to some degree. In view of this study content, I have the following suggestions:

1) Please clearly present the study motivation.

2) The consistency, confidence, RT, effort, and SoA should place in the left in the Figure 2-6.

3) Please unify the symbols. For example, p should use the orthography instead of italics.

4) Please improve the quality of Figure 7.

5) There is no in-depth discussion of the experimental results.

6) The authors should update "References" by including the latest publications from the Related Work.

7) Please check the paper carefully, there are many typos in the manuscript.

6. PLOS authors have the option to publish the peer review history of their article (what does this mean? ). If published, this will include your full peer review and any attached files.

**Do you want your identity to be public for this peer review?** For information about this choice, including consent withdrawal, please see our Privacy Policy .

Reviewer #1: No

Reviewer #2: No

Reviewer #3: No

---

## [Author Response · Author response to Decision Letter 1]

15 Jan 2025

Dr. Yuanchao Liu

Academic Editor

PLOS ONE

Dear Dr. Liu,

We are submitting our revision of PONE-D-24-57520, “Decision Makers Consider All Options in Choice Triplets” by Douglas Lee. We thank you for allowing us the opportunity to submit a revised version of our manuscript. We are very grateful for the feedback and suggestions provided by you and the reviewers, which have helped us to improve the manuscript. Below, we give an overview of the changes we made in response to specific comments by you and the reviewers:

Editor’s Comments

1) The study presents the results of original research in an appropriate fashion.

2) In the problem statement, the justification of conducting the research should be well explained.

We have added a paragraph at the end of the introduction to motivate this research:

In summary, the primary purpose of this study is to demonstrate that choice behavior is affected by all options within a choice set. Specifically, we seek to demonstrate that choices are facilitated (i.e., they are more consistent with value estimates, faster, and associated with feelings of greater confidence and lesser mental effort) when the differences in the value estimates of all available options are greater and when the average value across available options is greater. A secondary purpose of this study is to demonstrate that people revise their value estimates for all options during choice deliberation. The point is to show that all options are taken into consideration, not simply the best or second-best options.

3) Methodology and findings are performed to a high technical standard and sufficient detail, but the author should describe more about the sampling.

We now clarify that we recruited participants via the internal recruiting system of the Paris Brain Institute, without filtering for gender balance. We also clarify that the participants included 39 females and 11 males.

4) The research meets all applicable standards for the ethics of experimentation.

5) Conclusions are presented in a good way, but it needs a short general conclusion, mentioning some research limitations and implications.

We have now added a short conclusion section, including a subsection mentioning the limitations of the current study:

CONCLUSION

In summary, our results demonstrate that choice behavior is affected by all options within a choice set. Choices are facilitated by greater differences in the value estimates of all available options. They also seem to be facilitated when the average value across available options is greater. Apart from facilitation, our results support the idea that people revise their value estimates during choice deliberation. This value refinement occurs for all available options.

Limitations on Generalizability

This study examined choices between snack food options. It is possible that the results might have differed if different types of decisions were examined. The group of participants in this study was skewed towards the majority being female. We did not examine gender differences, but it is possible that the results might have differed if the group had gender balance or a male majority. In this study, we only considered a choice set size of three options. There might be a point at which adding more options to a set changes the result that all options are considered during deliberation. Our results did not replicate previous findings that value estimate certainty enhances choice consistency and confidence. It could be that metacognitive variables such as certainty become less impactful when choice set size increases beyond two options.

Reviewer 1 comments

The author(s) must reconsider to provide a comprehensive conceptual framework in the introduction that can help readers understand this topic better. The conceptual framework can also help the author(s) to refine the theory integration used in the research.

We have added a paragraph at the end of the introduction to motivate this research:

In summary, the primary purpose of this study is to demonstrate that choice behavior is affected by all options within a choice set. Specifically, we seek to demonstrate that choices are facilitated (i.e., they are more consistent with value estimates, faster, and associated with feelings of greater confidence and lesser mental effort) when the differences in the value estimates of all available options are greater and when the average value across available options is greater. A secondary purpose of this study is to demonstrate that people revise their value estimates for all options during choice deliberation. The point is to show that all options are taken into consideration, not simply the best or second-best options.

Reviewer 2 comments

In the last paragraph of the problem statement, the justification for conducting the research should be clearly and well stated.

We have added a paragraph at the end of the introduction to motivate this research:

In summary, the primary purpose of this study is to demonstrate that choice behavior is affected by all options within a choice set. Specifically, we seek to demonstrate that choices are facilitated (i.e., they are more consistent with value estimates, faster, and associated with feelings of greater confidence and lesser mental effort) when the differences in the value estimates of all available options are greater and when the average value across available options is greater. A secondary purpose of this study is to demonstrate that people revise their value estimates for all options during choice deliberation. The point is to show that all options are taken into consideration, not simply the best or second-best options.

In the methodology, it is stated that the sample size is 50, but 39 women are mentioned, and no explanation is given in this regard. And it is better to explain why women were chosen as the statistical sample.

We now clarify that the 50 participants consisted of 39 females and 11 males.

In the discussion section, it is better to add a brief general conclusion.

We have now added a brief general conclusion:

CONCLUSION

In summary, our results demonstrate that choice behavior is affected by all options within a choice set. Choices are facilitated by greater differences in the value estimates of all available options. They also seem to be facilitated when the average value across available options is greater. Apart from facilitation, our results support the idea that people revise their value estimates during choice deliberation. This value refinement occurs for all available options.

The limitations of the research, research suggestions and implications should also be discussed.

We have now added a subsection mentioning the limitations of the current study (in addition to the research suggestions that we previously included throughout the discussion section):

Limitations on Generalizability

This study examined choices between snack food options. It is possible that the results might have differed if different types of decisions were examined. The group of participants in this study was skewed towards the majority being female. We did not examine gender differences, but it is possible that the results might have differed if the group had gender balance or a male majority. In this study, we only considered a choice set size of three options. There might be a point at which adding more options to a set changes the result that all options are considered during deliberation. Our results did not replicate previous findings that value estimate certainty enhances choice consistency and confidence. It could be that metacognitive variables such as certainty become less impactful when choice set size increases beyond two options.

Reviewer 3 comments

The study work is meaning to some degree. In view of this study content, I have the following suggestions:

1) Please clearly present the study motivation.

We have added a paragraph at the end of the introduction to motivate this research:

In summary, the primary purpose of this study is to demonstrate that choice behavior is affected by all options within a choice set. Specifically, we seek to demonstrate that choices are facilitated (i.e., they are more consistent with value estimates, faster, and associated with feelings of greater confidence and lesser mental effort) when the differences in the value estimates of all available options are greater and when the average value across available options is greater. A secondary purpose of this study is to demonstrate that people revise their value estimates for all options during choice deliberation. The point is to show that all options are taken into consideration, not simply the best or second-best options.

2) The consistency, confidence, RT, effort, and SoA should place in the left in the Figure 2-6.

We have updated Figures 2-6 so that they are now in line with what we believe to be the suggestion of the reviewer.

3) Please unify the symbols. For example, p should use the orthography instead of italics.

We report p values in the format that is standard in our field, and the most common in scientific literature in general (we believe).

4) Please improve the quality of Figure 7.

The reviewer seems dissatisfied with the appearance of Figure 7 but does not say why. We are not sure what is dissatisfying about this figure, as it seems to clearly present the results in a simple and standard format.

5) There is no in-depth discussion of the experimental results.

We are not sure which additional discussion points the reviewer would prefer us to include. As it is, we have three full pages of discussion. This was a simple study with simple results, and we believe that our discussion section covers everything that we consider directly relevant.

6) The authors should update "References" by including the latest publications from the Related Work.

We are not sure which specific additional references the reviewer believes that we should include. To our knowledge, we have cited relevant papers appropriately.

7) Please check the paper carefully, there are many typos in the manuscript.

We have reviewed the manuscript many times and found no typos. We have now used Microsoft Word’s spelling and grammar checker and changed a few things that would be fine either way.

We hope that our efforts to respond to your concerns will be well received, and that our work will now be acceptable for publication in your journal.

Best Regards,

Douglas Lee

---

## [Decision Letter · Decision Letter 1]

24 Feb 2025

Decision Makers Consider All Options in Choice Triplets

PONE-D-24-57520R1

Dear Dr. Douglas G. Lee,

We’re pleased to inform you that your manuscript has been judged scientifically suitable for publication and will be formally accepted for publication once it meets all outstanding technical requirements.

Kind regards,

Yuanchao Liu

Academic Editor

PLOS ONE

Additional Editor Comments (optional):

Reviewers' comments:

Reviewer's Responses to Questions

**Comments to the Author**

1. If the authors have adequately addressed your comments raised in a previous round of review and you feel that this manuscript is now acceptable for publication, you may indicate that here to bypass the “Comments to the Author” section, enter your conflict of interest statement in the “Confidential to Editor” section, and submit your "Accept" recommendation.

Reviewer #2: All comments have been addressed

Reviewer #3: All comments have been addressed

2. Is the manuscript technically sound, and do the data support the conclusions?

Reviewer #2: Yes

Reviewer #3: Yes

3. Has the statistical analysis been performed appropriately and rigorously? 

Reviewer #2: Yes

Reviewer #3: Yes

4. Have the authors made all data underlying the findings in their manuscript fully available?

Reviewer #2: Yes

Reviewer #3: (No Response)

5. Is the manuscript presented in an intelligible fashion and written in standard English?

Reviewer #2: Yes

Reviewer #3: Yes

6. Review Comments to the Author

Reviewer #2: Dear author,

you made all the recommendations. Thanks for the convincing answers that were prepared in details.

Regards,

Reviewer #3: Authors have revised the manuscript according to my comments. Therefore, this manuscript can be accepted .

7. PLOS authors have the option to publish the peer review history of their article (what does this mean? ). If published, this will include your full peer review and any attached files.

**Do you want your identity to be public for this peer review?** For information about this choice, including consent withdrawal, please see our Privacy Policy .

Reviewer #2: No

Reviewer #3: No

---

## [Editor Report · Acceptance letter]

PONE-D-24-57520R1

PLOS ONE

Dear Dr. Lee,

I'm pleased to inform you that your manuscript has been deemed suitable for publication in PLOS ONE. Congratulations! Your manuscript is now being handed over to our production team.

Kind regards,

on behalf of

Dr. Yuanchao Liu

Academic Editor

PLOS ONE